# An Insight into the Medicinal Chemistry Perspective of Macrocyclic Derivatives with Antitumor Activity: A Systematic Review

**DOI:** 10.3390/molecules27092837

**Published:** 2022-04-29

**Authors:** Yan Liang, Ru Fang, Qiu Rao

**Affiliations:** Department of Pathology, Jinling Hospital, Nanjing University School of Medicine, Nanjing 210000, China; 15951826355@163.com

**Keywords:** macrocyclic compound, antitumor, synthetic methods

## Abstract

The profound pharmacological properties of macrocyclic compounds have led to their development as drugs. In conformationally pre-organized ring structures, the multiple functions and stereochemical complexity provided by the macrocycle result in high affinity and selectivity of protein targets while maintaining sufficient bioavailability to reach intracellular locations. Therefore, the construction of macrocycles is an ideal choice to solve the problem of “undruggable” targets. Inspection of 68 macrocyclic drugs on the market showed that 10 of them were used to treat cancer, but this structural class still has been poorly explored within drug discovery. This perspective considers the macrocyclic compounds used for anti-tumor with different targets, their advantages and disadvantages, and the various synthetic methods of them.

## 1. Introduction

Studies have shown that it is extremely difficult to develop small molecule drugs targeting proteins with extended binding sites, such as class B G-protein-coupled receptors (GPCRs), protein-protein interactions and some enzymes [1]. These difficulties are particularly acute in the development of antitumor drugs. The most commonly used “biological agents” for modulating these targets have some limitations, including high cost, reduced patient compliance, lack of cell permeability, and low oral bioavailability. Macrocyclic compounds have a degree of structural pre-organization that allows key functional groups to interact across extended binding sites in the protein, thereby reducing the major entropy loss during binding [2]. In order to successfully bind to a target protein, a molecule has to adopt a bioactive conformation. There is a lower entropic cost when the ligand binds with the protein by limiting the number of conformations available to the unbound molecule. Macrocycles have restricted internal bond rotations and are conformationally constrained, although not completely rigid. Macrocycles are potentially adaptive molecules with enough flexibility to efficiently interact with flexible binding sites in proteins, at the same time minimizing the internal entropy penalty associated with the change from the unbound to the bound state of the ligand. Reduction in the overall motion of a receptor, although an unfavorable entropic change, may increase the strength of intermolecular interactions to a ligand, such as hydrogen bonds, thus increasing favorable enthalpic contribution. These characteristics of macrocyclic compounds make “molecular macrocyclization” a key means to solve the above problems [3].

Cyclization of a linear molecule into a macrocyclic ring constitutes a significant change in molecular shape, biological activity, and drug-like properties. Compared with an acyclic linear molecule, cyclized molecules have better physicochemical properties, such as good solubility, lipophilicity, metabolic stability, bioavailability and overall pharmacokinetics [4].

Macrocyclic compounds are not only larger versions of small molecules but can also be considered the smallest biomolecule capable of chimeric targets [5]. Although macrocyclic compounds have been shown to have therapeutic potential, they have not been fully researched and exploited in drug discovery. Most of the macrocyclic drugs currently on the market are natural products with complex structures. The complex structure increases the difficulty of synthesis and the cost of production, leading the pharmaceutical industry to be cautious about the development of macrocyclic drugs [6].

However, significant progress has recently been made to simplify the synthesis of macrocyclic compounds. As described in this paper, some research groups have synthesized macrocyclic compounds that are unrelated to natural products and have demonstrated that these compounds might improve drug affinity and selectivity while maintaining acceptable drug-like properties [7].

Some preclinical or clinical studies have improved our understanding of the absorption, distribution, metabolism and excretion (ADME) of macrocyclic compounds. Currently, it is difficult to obtain small molecules with good potency and overall pharmacokinetics during antitumor drug development [8]. Therefore, the interactions between macrocycles and their target proteins should continue to be explored [9]. In this paper, based on different target proteins, the representative macrocyclic compounds used for the treatment of malignant tumors are reviewed, and the unique interactions of key macrocyclic compounds with their target proteins and the synthesis process are introduced in detail. It is hoped that this review will provide guidance for the design and development of macrocyclic compounds.

## 2. Macrocyclic Compounds for the Treatment of Malignant Tumors

Currently, the design strategy of cyclization is only applied to the development of a few antitumor drugs, and different design ideas have been adopted for the development of macrocyclic compounds for different target proteins. A summary of different strategies is described in this section.

### 2.1. Anaplastic Lymphoma Kinase (ALK) and C-Ros Oncogene 1 (ROS1) Macrocyclic Inhibitors

Crizotinib (**1**) is an ATP-competitive, multi-targeted protein kinase inhibitor developed by Pfizer that inhibits Met/ALK/ROS. It has been clinically confirmed that Crizotinib has significant efficacy in tumor patients with abnormal ALK, ROS and Met kinase activities, respectively. It was approved by the U.S. Food and Drug Administration (FDA) in 2011 as the first ALK inhibitor as a first-line treatment for ALK-positive lung cancer. Although Crizotinib initially showed good efficacy in ALK-positive cancers, patients eventually developed resistance, which was caused by point mutations in the ALK gene (the gatekeeper mutation frequency exceeds 40%) [10]. Additionally, patients with brain metastases have a poor prognosis with a median survival of 2.5 months, which may be partly due to the poor blood-brain barrier (BBB) permeability of Crizotinib. Therefore, the next generation of ALK inhibitors must have adequate CNS exposure and broad-spectrum ALK potency.

Lipophilic efficiency (LipE = pK_i_ (or pIC_50_) − logD) is a critical parameter to meet the objective of avoiding transporter-mediated efflux at the BBB and cancer cell surface. Molecular weight (MW) is inversely related to permeability, and smaller ligands are more likely to be more effective inhibitors because they have a greater chance to achieve the desired CNS exposure. MDR BA/AB is the ratio of basolateral-to apical/apical-to-basolateral flux in Madin-Darby canine kidney (MDCK) cells transfected with the multidrug-resistance gene (MDR) that encodes for human P-glycoprotein 1 (Pgp 1). The MDR BA/AB ratio (>2.5) represents P-glycoprotein 1 (Pgp 1) efflux, which is consistent with CNS exclusion [10]. In general, the design of new inhibitors requires a novel approach to solving the current difficulties.

The cocrystal structure of the wild-type kinase domain of ALK with **2a** (Figure 1A) showed that the aminopyridine core interacted with the hinge portion of the protein via an acceptor-donor motif. The substituted fluorophenyl headgroup is very close to the heteroaromatic tail, the distance being 4.16 Å (Figure 1B), so it is possible to form macrocyclic compounds by linking the two groups from the triazole 4-carbon to the methoxy carbon of **2a** [10]. These templates were expected to reinforce the binding conformation and provide additional protein ligand interactions in the linker region. In addition, because the macrocyclic analog is more compact and has a reduced surface area, it may provide a permeability advantage. Solvent-accessible surface areas (SASAs) for macrocyclic molecules are almost 10% smaller than those for non-macrocycles within defined molecular weight ranges. Replacing molecular weight with SASA may provide a more accurate descriptor for molecular size and increase the probabilistic assessment in the future design of macrocyclic inhibitors. Driven by LipE and MW optimization, researchers have designed a series of macrocyclic inhibitors to achieve the desired CNS ADME properties while maintaining efficacy (Table 1).

A series of 12–14-membered ether-linked macrocyclic compounds were first prepared to verify the design of the macrocycle (Figure 1C). The summarized data shows that the smaller ring size always provides the most lipophilic efficient macrocycles, and the most efficient analogue **2b** displayed the highest LipE (4.4) and excellent cellular potencies (pALK IC_50_ = 1.0 nM; pALK-L1196M IC_50_ = 20 nM) [10]. However, since macrocyclic ethers were generally too lipophilic, this series of derivatives still lacked the required efficiency for more facile overlap of potency, ADME, and CNS availability. Subsequently, a set of amide-linked macrocycles was prepared, which attempted to reduce lipophilicity while retaining LipE with a lactam linker. Compound **2c** showed similar cellular potencies against ALK and ALK-L1196M to those of compound **2b**, but the lipophilicity of **2c** was reduced by one unit, and LipE was increased by two units. Additionally, amide macrocycles exhibited low clearance and low MDR BA/AB efflux ratios and achieved the desire CNS ADME properties while maintaining efficacy [10].

Next, a study of selectivity against other kinases was conducted by targeting residues which were specific in ALK protein. Leu1198 in ALK protein is relatively conserved in 26% of kinases. In the same position as other kinases, most of them are Phe/Tyr. The difference in steric hindrance between Leu and Phe/Tyr can be used to design small molecule compounds that specifically target Leu but not Phe/Tyr, to achieve high selectivity for ALK protein. The TrkB protein containing Tyr635 at the position of ALK Leu1198 was selected as the research object. Compound **2c** showed strong inhibitory activity on TrkB with the IC_50_ value being 0.5 nM [10]. The ortho-methyl group in pyrazole of **2c** is far away from TrkB Tyr635. The distance is 2.62–3.91 Å between methyl and the terminal hydroxyl group (OH), terminal carbon (CZ), and adjacent carbon (CE) of Tyr635 (Figure 2A). Fortunately, compound **2d** with cyano group replacing methyl group of **2c** showed decreased TrkB inhibitory activity, with an IC_50_ of 23 nM and the selectivity of ALK over TrkB was 38 times. Compared with compound **2c**, the distance between the cyano group of **2d** and the terminal atoms of Tyr635 was shorter, at 1.33–2.89 Å and might have a potential clash with Tyr635 in TrkB (Figure 2B). Moreover, an unfavorable desolvation penalty and electrostatic repulsion between the electron-rich nitrile nitrogen atom and Tyr635 might further contribute to selectivity. In general, the introduction of different lengths and different sterically hindered substituents in the ortho position of the ALK ligand pyrazole could achieve various inhibitory activities against the TrkB protein to achieve selectivity.

Macrocyclic compound **2d** exhibited significant inhibitory activity against target proteins including ROS1, wild type and clinical mutants of ALK, and high selectivity for various kinases (Figure 2C,D). In addition, **2d** had suitable lipophilic efficiency (LipE = 5.4), excellent membrane permeability (MDR BA/AB ratio = 1.5), good metabolic stability (HLM Cl < 8), and very efficient oral bioavailability (F% = 100%). More importantly, compound **2d** could also achieve good CNS availability (Figure 2E), thereby having a good therapeutic effect on brain metastases. Compound **2d** (PF-06463922, Lorlatinib), as a highly potent and selective macrocyclic ALK/ROS1 inhibitor, has been successfully approved for the treatment of nonsmall cell lung cancer NSCLC by FDA in 2018, which is a typical example in the development of macrocyclic drugs [10].

### 2.2. Janus Kinase 2 (JAK2) Macrocyclic Inhibitors

The Janus kinase family (JAK1, JAK2, JAK3 and TYK2) are intracellular non-receptor tyrosine kinases that play important roles in the control of cell proliferation, cell differentiation and survival [11,12]. Clinically, blood-related diseases such as leukemia and myelofibrosis, as well as autoimmune diseases such as rheumatoid arthritis and ankylosing spondylitis, are closely related to JAK-mediated signaling pathways. For example, JAK2 inhibitors are mainly used for myeloproliferative neoplasms (MPNs) and clinical treatment of lymphoma. Based on highly selective JAK2 inhibitors, macrocyclic inhibitors with different inhibitory activities to FLT3 or CDK2 were developed through structural cyclization, which can be clinically used to treat different indications. The macrocyclic compound **3c** has a significant selective inhibitory effect on JAK2 and FLT3, while it has no obvious inhibitory activity on JAK1, JAK3, and CDK2. It is mainly used to treat myelofibrosis and lymphoma. Another compound **3d**, which has a significant selective inhibitory effect on JAK2, FLT3, and CDK2 at the same time, is mainly used for the clinical treatment of leukemia and myeloma. The macrocyclic compound **3e** has significant inhibitory activities on JAK2, FLT3 and TYK2, but its inhibitory activities on JAK1, JAK3, and CDK2 were significantly weaker and is mainly used to treat internal rheumatoid arthritis. In the following, the medicinal chemistry modification work of these three macrocyclic compounds is briefly summarized.

First, screening by an internal library showed that compound **3a** has extensive kinase inhibition and has reasonable activity in the cell lines tested [11]. However, since the pyrimidine core scaffold has been heavily patented without the potential to develop proprietary compounds, researchers speculated the formation of macrocycles by connecting the open ends of **3a** to develop a novel macrocyclic scaffold molecule, without infringing patents, and preserve the binding pattern with the kinase hinge region (Figure 3).

Researchers first started with a suitable linker, hoping to introduce some hydrophilic, oxygen-containing linking chains, so as to ensure proper solubility and metabolic stability. Compound **3b** has a linker of eight atoms in length and two benzyl ethers, showing good activity and selectivity. However, compound **3b** has a higher cLogP value and poor water solubility. Molecular docking studies have found that an oxygen atom of benzyl ether forms a new hydrogen bond interaction with Ser936, which is the reason for the high inhibitory activity of **3b** on JAK2. In addition, the methoxy group on the C ring directly points to the solvent channel, which provides a chemical modification space for introducing basic groups to improve water solubility. In addition, the researchers systematically investigated the substituents on the A, B, and C rings. Finally, compound **3c**, the R_1_ substituent on the C ring was substituted with a pyrrolidine carbon chain, showing the best activity, selectivity, and physical and chemical properties [11].

Compound **3c** showed potent inhibitory activity on JAK2 and FLT3, with IC_50_ of 23 nM and 22 nM, respectively, and also had selectivity for CDK2, JAK1 and JAK3 (Figure 4) [12]. In the BaF3 cell line, **3c** also showed significant inhibitory activity. It has good water solubility (150 ug/mL), no inhibitory activity on CYP3A4 and CYP2D6, moderate liver microsomal stability, and good oral bioavailability (mice, dogs, rats) [12].

In an in vivo efficacy test, **3c** also showed excellent anti-tumor activity. Compound **3c** was therefore chosen for the evaluation of two mouse tumor models: Ba/F3-JAK2V617F and MV4-11 allograft and xenograft studies representing cell lines dependent on mutant JAK2 and FLT3 signaling, respectively [11]. In the Ba/F3-JAK2V617F mouse model, control mice showed splenomegaly and hepatomegaly (~7-fold and 1.6-fold, respectively). At the end of the study, **3c** treatment of 150 mg/kg p.o. b.i.d. significantly improved disease symptoms, with 42% normalization of spleen weight and 99% normalization of liver weight [11]. The MV4-11 xenograft model was established in nude mice, tumor growth was measured up to 55 days, and median time-to-end point (median survival) was calculated at the end of treatment. Kaplan-Meier survival curves for different groups showed that the median survival of the vehicle group was 33.0 days, and the 50 mg/kg and 100 mg/kg **3c** treatment groups exhibited a significantly increased median survival of 55 days (*p* < 0.01) with no significant weight loss in the treated animals at the end of the study [11].

In general, researchers have discovered a series of small macrocyclic compounds as potent inhibitors of JAK2 and FLT3 kinases, and a subset of diverse compounds were selective against JAK family and CDK2. Compound **3c** showed significant survival benefit at very well-tolerated doses, and in two completed Phase 2 trials, it has shown clinical benefits in patients with myelofibrosis and lymphoma.

In another report, the researchers discovered the clinical candidate SB1317/TG02 (**3d**) (Figure 5) through a similar macrocyclization design and a series of molecular docking studies [12]. **3d** is a potent inhibitor of cyclin dependent kinases (CDKs), Janus kinase 2 (JAK2), and Fms-like tyrosine kinase-3 (FLT3). Cyclin-dependent kinases (CDKs) are threonine kinases that play important roles in cell cycle control (CDK1, 2, 4, and 6), transcription initiation (CDK7 and 9), and neuronal function (CDK5). Unlike compound **3c**, **3d** has no selectivity for CDKs. The docking model indicates that this is due to the fact that the basic nitrogen contained in the macrocycles’ linker forms a salt bridge with Asp86 in CDK2 and Asp698 in FLT3, while the Asp86 residue is conserved in most CDKs, resulting in potent inhibition of pan-CDK of these compounds. Due to the presence of basic nitrogen in the macrocyclic linker of **3d**, it is not necessary to introduce polar solubilizing side chains to improve solubility.

After obtaining **3d** by stepwise structural modification, a series of biological evaluations were performed. Compound **3d** exhibited potent inhibitory against CDK2, JAK2, and FLT3, with IC_50_ values of 0.013, 0.073, and 0.056 nM, respectively [12]. In the cell proliferation inhibition assay, **3d** was potent against all the cell lines tested including HL-60, RAMOS and solid tumor cell lines such as colon (HCT-116, COLO205) and prostate (DU145). Additionally, **3d** also has acceptable pharmacokinetic properties, which was further used for evaluation in human tumor xenograft mouse models. Based on their relevance in cancer, HCT-116 colon cancer and Ramos B-cell lymphoma models were selected. In the colon cancer mouse model, treatment with **3d** at a dose of 75 mg/kg po three times per week significantly inhibited the growth of tumors with a mean TGI of 82% [12]. Two different dosing regimens of **3d** were explored in the MV4-11 xenograft model: 75 mg/kg po q.d. on a 2 days on and 5 days off schedule and 15 mg/kg ip q.d. on a 5 days on 5 days off schedule were started 12 days after cell inoculation for 15 days. Compared with the control group, the mean TGIs of the oral and ip delivery methods were 42% and 63%, respectively, and significantly inhibited tumor growth. On the basis of its favorable pharmaceutical and pharmacological profile, **3d** is currently being evaluated in phase 1 clinical trials in leukemia patients [12].

### 2.3. Discovery of a Next Generation of EGFR Tyrosine Kinase Inhibitors

The epidermal growth factor receptor (EGFR) is a receptor tyrosine kinase that transduces mitogenic signals and acts as an oncogenic driver in a subset of lung tumors when carrying an activating mutation like short in-frame deletions in exon 19 (del19) or L858R, a single missense mutation in exon 21 [13]. Although tyrosine kinase inhibitors (TKIs) can shrink tumors significantly, the apparent tumor shrinkage response of tumors to TKIs is usually not durable. Due to the additional T790M and C797S mutations in the EGFR kinase domain that confer TKI resistance, most patients relapse within two years of therapy. Regrettably, both resistance mutations cannot be inhibited by currently approved EGFR TKIs.

In 2019, Engelhardt et al. discovered a series of macrocyclic EGFR TKIs which potently inhibit the above-described EGFR variants [13]. Researchers started a screening program based on high selectivity for mutated EGFR proteins over EGFR^wt^ protein to obtain hit compounds. After compound **4a** (Figure 6) was obtained, an N atom was introduced to the aromatic ring A to form a hydrogen bond interaction with Thr854. In order to balance the relationship between selectivity and activity, modifications on the aromatic ring B were conducted to find the appropriate dihedral angle between the two aromatic rings and to form a suitable interaction with Lys745 (Figure 7A). Subsequently, compound **4b** possessing good selectivity and activity was obtained. The active and inactive conformational states of compound **4b** were quickly obtained based on the computational conformational analysis.

In order to confine the inhibitor to the active binding conformation of compound **4b**, a series of macrocyclic variants with different linker lengths and connection points were synthesized (Figure 6). Among inhibitors, the better macrocyclic ligand **4c** showed a good overlap with the “active” conformation of **4b** (Figure 7B). The X-ray structure of compound **4c** in complex with EGFR showed that there are no new interactions formed, so improving potency was caused by the conformational restrictions.

In order to find better inhibitors, follow-up transformation research of **4c** has focused on the following four aspects: (1) implementing a polar interaction with Lys745 by adjusting the ring B; (2) maintaining selectivity by keeping the methyl group in the R_1_-position; (3) restricting the ligand to only one enantiotopic minimum conformer by installing a stereo center in the linker; (4) improving the DMPK parameters by introducing the solubilizing groups at the R_3_-position.

Compound BI-4020 (**4d**) is the most promising ligand, with its anti-proliferative ability of BF3 cells reaching the sub-nanomolar level and having better selectivity. Studies have shown that compound **4d** can cause tumor regression in the Osimertinib-resistant PC-9 (EGFR^del19 T790M C797S^) triple mutant NSCLC xenograft model in mice. The excellent in vitro and in vivo activity of **4d** proves its potential as a clinical compound [13].

### 2.4. Discovery of Novel Pyridones and Pyridone Macrocycles as Potent Bromodomain and Extraterminal Domain (BET) Family Bromodomain Inhibitors

In 2017, Wang et al. disclosed novel pyridones and pyridone macrocycles as potent bromodomain and extra-terminal domain (BET) family bromodomain inhibitors [14]. The Bromodomain and Extraterminal (BET) family is a subset of related bromodomain-containing proteins that includes BRD2, BRD3, BRD4, and BRDT. Then, compound **5a** was discovered through the protein-based NMR fragment screening against the second bromodomain of BRD4 (BRD4-BDII). **5a** showed a strong spectral shift in the NMR screen, but it was a weak BRD4 inhibitor with a K_i_ of 160 μM [14].

Based on the X-ray co-crystal structure of compound **5a** with BRD4 BDII, the researchers performed a series of structural modifications on compound **5a** to obtain compounds **5b** (BRD4 K_i_ = 58 nM) and **5c** (BRD4 K_i_ = 13 nM) [14]. The X-ray co-crystal structure of pyridone **5b** (Figure 8) showed that spatial orientation of the aliphatic ether side chain on the pyridone ring is in close proximity to the ortho carbon position of the phenyl C-ring. It was hypothesized that cyclic systems formed by tethering these sites might result in improved potency by reinforcement of a low energy binding conformation to the BRD protein and by reduction in the number of rotatable bonds.

Subsequently, the researchers determined that the optimal ring size was a five-atom linker (compound **5d**, K_i_ = 1.5 nM), and the candidate compound **5e** (Figure 8) (BRD4 K_i_ = 8.9 nM) was obtained by incorporating a halogen atom on the C-ring. Compound **5e** showed the best oral exposure in mice with low clearance (Clp = 14 L/h/kg), outstanding absorption (FaFg = ~1), and excellent oral bioavailability (F = 95%) among these compounds. In an in vivo efficacy experiment, compound **5e** exhibited excellent TGI of 85% with the dose of 2 mg/kg [14].

In summary, macrocyclic inhibitor **5e** exhibits somewhat superior antitumor activity compared to acyclic pyridone **5c**, which could be caused by the increased rigidity of the desired binding conformation along with increased van der Waals interactions.

### 2.5. Discovery of Pyrazolo [1,5-a]pyrimidine B-Cell Lymphoma 6 (BCL6) Binders and Optimization to High Affinity Macrocyclic Inhibitors

Inhibition of the protein-protein interaction between B-cell lymphoma 6 (BCL6) and corepressors has been implicated as a therapeutic target in diffuse large B-cell lymphoma (DLBCL) cancers [15]. In 2017, McCoull et al. identified a pyrazolo [1,5-a]pyrimidine series of BCL6 binders (**6a** and **6b**) from a fragment screen in parallel with a virtual screen.

As is shown in Figure 9, compound **6c** (BCL6 FRET IC_50_ = 100 μM) was obtained by modifying the 5-position of **6b**, which bound Arg24 through an ionic interaction with higher affinity [15]. In order to form both van der Waals and hydrogen bonding interactions with Glu115, the 7-cyclopropylamino group was replaced by a bicyclic lactam group to obtain compound **6d** (BCL6 FRET IC_50_ = 0.35 μM).

The cocrystal structure of **6d** and BCL6 (Figure 9B) showed that the 3-position carbon of the pyrrolidine ring and the aniline are 7 Å apart. The fact that the bound conformation of **6d** is not the lowest energy conformation of the isolated ligand, suggests that the construction of macrocyclic compounds could force the bioactive conformation of binding to BCL6 to be a more prevalent conformation. The introduction of a bis-ether linker of six atoms obtained the derivative **6e** (BCL6 FRET IC_50_ = 0.0029 μM) [15].

A remarkably close overlay between the acyclic **6d** and macrocyclic **6e** (Figure 9C) exhibited that the atoms in common between **6e** and **6d** overlay almost perfectly (RMSD = 0.96 Å). Finally, the selectivity improve against off-target kinase CK2 was conducted, and the potency over a number of DLBCL lines and MM cell line was assessed. However, the researchers only observed weak antiproliferative effects and found no differences between BCL6-sensitive or insensitive cell lines. Therefore, whether BCL6 is an effective target for the treatment of DLBCL remains to be confirmed.

### 2.6. Design of Macrocyclic Hsp90 Inhibitors with Increased Metabolic Stability and Potent Cell-Proliferation Activity

Heat shock protein 90 (Hsp90) is a constitutive and ubiquitously expressed ATP-dependent molecular chaperone in mammalian cells. Hsp90 controls the conformational maturity, stability and function of its substrate protein, called the client protein. It is well known that client proteins play an important role in proliferation, invasion, survival, metastasis and angiogenesis of cancer cells. Thus, Hsp90 inhibitors can effectively inhibit tumor growth and progression by degrading client proteins [16]. Different ATP competitive chemical types have evolved into effective N-terminal Hsp90 inhibitors, some of which have entered clinical trials [17].

The first Hsp90 inhibitor is the macrocyclic natural product Geldanamycin (**7**) (Figure 10), belonging to the class of ansamycin [18]. The structural modification of this natural product usually has the characteristics of poor solubility and narrow therapeutic window. Subsequently, more and more small molecule inhibitors entered the human field of vision. NVP-AUY922 (**8**), which was subjected to clinic trial in 2007, is considered a potential new Hsp90 inhibitor [19,20]. In recent years, Serenex et al. has been dedicated to the discovery of Hsp90 inhibitors [21], and their research ultimately led to a series of 2-aminobenzamides exhibiting low nanomolar potency in proliferation assays [22]. Among the reported compounds, the glycine pro-drug SNX-5422 (**9**) was forwarded to clinical trials.

In order to seek for effective and novel small molecule inhibitors of Hsp90, researchers studied the X-ray crystallography of structural analog of **9**. Simple molecular modeling showed that these compounds form two key hydrogen bonds with Hsp90: benzamide interacting with Asp93, and the carbonyl group on the tetrahydrofuranone moiety interacting with Tyr139. The optimal dihedral angle of the aromatic moiety allows the two hydrogen bonds to be accessible.

On this basis, they designed a unique macrocyclic ortho-aminobenzamides and changed the ring size, substituents and stereo chirality. Finally, a series of macrocyclic compounds with good solubility and microsomal stability were obtained, which showed effective inhibition of Hsp90 in enzyme and cell proliferation assays. Compound **10a** (Figure 11) was one of the most potent compounds in the series (Hsp90 enzyme IC_50_ = 0.11 μM, HCT116 proliferation EC_50_ = 0.09 μM) showing high water solubility (>100 μg/mL) and stability in rat microsomes (t_1/2_ > 30 min). Similarly, analog **10b** also had high effectiveness (Hsp90 enzyme IC_50_ = 0.09 μM, HCT116 proliferation EC_50_ = 0.03 μM) and solubility (>100 μg/mL), whereas its rat microsome stability was moderate (t_1/2_ = 15 min). The team further synthesized the macrocyclic compound **10c**, which had acceptable solubility and rat microsomal stability on the basis of maintaining the original enzymatic and cellular activities (Hsp90 enzyme IC_50_ = 0.09 μM, HCT116 proliferation EC_50_ = 0.04 μM). Unfortunately, compound **10c** is potent inhibitor of the hERG ion channel due to their presence of basic amines (IC_50_ = 3.2 μM).

The X-ray structure of **10a** shows a second water molecule proximal to the benzamide interacting with Ser52, Ile91, Asp93 and Thr184 representing the dense network of conserved water molecules characteristic for the N-terminal ATP-binding site of Hsp90. This phenomenon confirms that the macrocyclic tether linking the aminobenzamide to the tetrahydroindolone is, indeed, oriented toward the solvent exposed region of the binding site. In a subsequent study, the authors designed and synthesized a series of macrocyclic amide **10d** and lactam **10e**–**10f** (Figure 12) to mitigate hERG activity by further functionalization and derivatization of the tether [23].

Among the lactams, the analog **10g** retains biological activity and physical properties while improving computational properties such as TPSA and clogP. More important, compound **10g** showed only about 7% inhibition of channel hERG at 11 μM, which was significantly improved compared to compound **10c**, which inhibited hERG by up to 78% at the same concentration. In addition, **10g** has significant anti-proliferative activity against various other cell lines.

### 2.7. Structure-Based Design, Synthesis, and Biological Evaluation of Potent and Selective Macrocyclic Checkpoint Kinase 1 Inhibitors

Checkpoint kinase 1 (Chk 1), a serine/threonine protein kinase, is a key mediator in the DNA damage-induced checkpoint network [24,25,26]. Its mechanism of action is mainly to activate the key checkpoints (S, G_2_) of mitosis of tumor cells and repair damaged tumor cells. Inhibition of ChK1 can make tumor cells sensitive to various DNA damaging agents and cannot perform DNA repair [27,28,29]. The inhibition of Chk1 may significantly improve the efficacy and selectivity of DNA-damaging agents in the clinic [30,31,32,33]. Therefore, many classes of Chk1 inhibitors have been reported. In 2007, Tao et al. discovered a series of novel macrocyclic inhibitors based on the urea scaffold [34].

Firstly, two potent and selective urea-based Chk1 inhibitors, **11a** and **11b**, were selected as lead compounds (Figure 13) [35,36]. Subsequently, the modification of the C6′-position in the pyrazinyl ring resulted in compounds **11c** (IC_50_ = 26 nM) and **11d** (IC_50_ = 22 nM). These two compounds can abrogate the cell cycle checkpoints and significantly potentiate the cytotoxicity of topoisomerase inhibitors.

An X-ray cocrystal structure of a Chk1-**11d** complex (Figure 14A) showed the substituents at the C6′- and C2-positions are essentially proximal, and the urea amide bonds were in one cis and one trans orientation. Therefore, researchers have tried to restrict the conformation by connecting terminal substituents to improve potency. A series of macrocyclic compounds were obtained by olefin metathesis reaction.

Structure-activity relationship studies demonstrated that the size of the ring has only modest effect on the inhibitory activity. The X-ray cocrystal structure of **11e** (Figure 14B) showed that the carbon chain makes favorable van der Waals contacts with Glu91, Asn135, Leu137, Glu134, and Ser147. Compounds **11e** and **11f** showed considerable Chk1 inhibitory activity in vitro, but the introduction of nitrile group brought better cellular activity to compound **11f**. Moreover, examination of the X-ray cocrystal structure of **11e**-Chk1 complex shows that the 4-position of the phenyl ring can tolerate a wide variety of substituents. The researchers designed ether, amino, dialkylamino, alkynol and aromatic ring analogues under the premise of nitrile group retention.

Compounds with the advantage of enzyme inhibitory activity were screened for anti-tumor proliferation activity in HeLa cells of the human cervical cancer cell line with a p53 deletion. As shown in Table 2, the EC_50_ values of these compounds were measured alone or in the presence of 150 nM doxorubicin (Dox), which is a clinical topoisomerase II inhibitor that can block HeLa cells G_2_/M checkpoint. **11g**, the most effective compound, showed little antiproliferative activity when used alone, but when used in combination with Dox, it significantly enhanced the cytotoxicity of the DNA-damaging agents Dox.

### 2.8. Non-Natural Macrocyclic Inhibitors of Histone Deacetylases (HDACs): Design, Synthesis, and Activity

Histone deacetylase (HDAC) changes the accessibility of transcription factors to DNA by controlling the acetylation status of histone lysine residues, thereby affecting the chromatin remodeling process. This kind of epigenetic therapy, which does not involve changes in DNA sequence, is a rapidly developing new field in pharmacology [37,38]. Researchers have proven that the inhibition of HDACs have an antiproliferative effect on tumor cell lines [39]. Early reported compound **12** is an unnatural class I and II HDAC inhibitor. The ω-benzyloxy-substituted suberoyl-based hydroxamic acid **13a** (Figure 15) was discovered by Auzzas et al., in 2007 [40].

As shown in Table 3 and Table 4, compared with compound **12**, **13a** has the same inhibitory effect on HDAC1-11, but it shows better anti-proliferation effect than **12** on two different human cancer cell lines lung cancer H460 (IC_50_ = 0.52 µM) and colon cancer HCT-116 (IC_50_ = 0.22 µM) [41,42]. Subsequently, a series of non-peptide chiral macrocyclic compounds were designed and synthesized by using compound **13a** as the mother nucleus for structural modification and adopting the conformational concept of macrocycles.

Based on the suberyl group in the core of **13a**, a lipophilic linker was selected to form a branch point at the α-position of the suberic acid chain in the space-limited template, and the structure was modified to design and synthesize the macrocyclic compound **13b** which has a maximum common substructure required to target the whole HDAC pane. It shows sub-nanomolar activity (IC_50_ = 0.84 nM) against HDAC6, and has good cytotoxic activity against lung cancer (IC_50_ = 1.05 µM) and colon cancer cell lines (IC_50_ = 0.69 µM) [42].

The predicted binding mode analysis of **13b** in HDAC8 revealed the possibility of adding an additional contact point with a conserved Asp101 residue on the edge of the isozyme by appropriately introducing the hydrogen donor group in the macrocycle. On this basis, the macrocyclic hydroxamic acid **13c** and **13d** were synthesized and evaluated. Compound **13d** increased the inhibitory activity of HDAC6 (IC_50_ = 0.40 nM), but the in vitro anti-tumor proliferation activity of lung cancer (IC_50_ = 6.80 µM) and colon cancer cell lines (IC_50_ = 3.74 µM) was significantly reduced [42]. Compound **13c** showed relatively excellent inhibitory ability at both the molecular level and the cell level. Its inhibitory activity on HDAC6 and the in vitro anti-tumor proliferation activity of cells remained at the nanomolar level, becoming a new candidate compound for the next compound design provides the basis.

### 2.9. Structure-Based Design and Synthesis of Novel Macrocyclic Pyrazolo[1,5-a][1,3,5]triazine Compounds as Potent Inhibitors of Protein Kinase CK2

Protein kinase CK2, a highly conserved and pleiotropic serine/threonine kinase, plays an important role in cell growth, proliferation, and survival. Many studies support the rationale of inhibiting CK2 as a potential approach for cancer therapy. In 2008, Nie et al. disclosed a novel class of macrocyclic pyrazolo[1,5-a][1,3,5]triazines as potent CK2 inhibitors [43].

Compound **14a** (Figure 16A) was first defined as a potent CK2 inhibitor, but it exhibited weak cellular activity against prostate and colon cancer cell lines. The poor cellular activity may be related to the near-plane molecular structure of compound **14a**, which results in poor membrane permeability. As shown in Figure 16B, the cis-acetamide group of compound **14a** forms two strong hydrogen bond interactions with the protein residues Asp175 and Lys68 of CK2, resulting in the right phenyl group and its acetamide structure being buried in the hydrophobic pocket. In this way, the permeability of the membrane is poor, which in turn affects its cell viability. The researchers observed that the methyl group of the acetamide on the right was only about 4 Å apart from the nitrogen atom of the C8-nitrile group. Therefore, appropriately extending the steric alkyl chain can link C8 and acetamide to form a macrocyclic system. Therefore, the macrocyclic compound **14b** was designed to form a bulge out from the molecule and change the planar characteristics of non-cyclized pyrazolo triazines to improve their cell membrane permeability. The co-crystal structure of compound **14b** (Figure 16C) revealed that the alkyl linker of the macrocyclic derivative fits quite elegantly into the hydrophobic cavity and the amide group still maintains two hydrogen bonds with Asp175 and Lys68. Even though compound **14b** lost 100-fold enzyme inhibitory potency compared with **14a**, it showed effective cellular potency against prostate and colon cancer cell lines.

Encouraged by the improvement of cell activity, based on the available co-crystal structure, another series of macrocyclic pyrazolo triazine derivatives (**14c**–**14j**) was designed and synthesized (Table 5). Compared with the structure of compound **14b**, they retain the aniline group on the left side of **14a** and introduces different substituents at the meta position of aniline to point to the solvent exposed area. Therefore, the incorporation of m-amide groups may enhance their enzyme inhibitory efficacy and increase solubility. The subsequent evaluation results also verified these conjectures. Compounds **14c**–**14j** have been further improved at the level of enzyme activity and cell activity. At present, research on these types of compounds is still ongoing.

### 2.10. Structure-Based Drug Design of a Highly Potent CDK1, 2, 4, 6 Inhibitor with Novel Macrocyclic Quinoxalin-2-One Structure

The cyclin-dependent kinase (CDK) protein family plays a key role in cell-cycle regulation in eukaryotic cells [44]. CDK1, 2, 4 and 6 are attractive targets for new anticancer drugs [45]. Based on the diarylurea class of inhibitor **15a**, Kawanishi et al. discovered a series of macrocyclic CDK1,2,4,6 inhibitors in 2006 [46].

At first, compound **15b** (Figure 17) was obtained by converting a linear system to a more coplanar ring, which showed moderate CDK4-inhibitory activity. For better potency, the isoindol-1-one group in compound **15c** was optimized to provide a more favorable coplanar conformation, but its cellular potency needed to be improved (IC_50_ = 310 nM; E2F assay). Subsequently, compound **15d** with a 5-carbon linker was designed, which showed the best activity (IC_50_ = 30 nM; E2F assay). The chains of carbon atoms linked the quinoxaline-2-one and the benzoisothiazol-1-one, fixed the dihedral angle and filled the void space around the ribose site of the ATP-binding pocket.

Compound **15e** was obtained by inserting amines into the linker of compound **15d**, which is insufficiently soluble in water. Therefore, more hydrophilic compounds, such as **15f**, were designed. Then, the pyrrolidine ring was introduced into the linker to obtain compound **15g**, which further improved the cell efficiency. The X-ray structure of the eutectic complex with CDK2 reveals that compound **15g** forms hydrogen bonds with Val83, additional hydrogen bonds with Lys33 in the salt-bridge region and hydrophobic interactions with Ala144, Leu134, and Asn132 were also formed.

In conclusion, compound **15g** showed best potency in both enzyme and cellular assay, which is available for IV administration and showed potent CDK-inhibitory activity in a preclinical animal model.

### 2.11. Macrocyclic Pyrrolobenzodiazepine Dimers as Antibody-Drug Conjugate Payloads

Antibody-drug conjugates (ADCs) can harness the high specificity of antibodies to directly target small molecule drugs to tumor cells. Pyrrole benzodiazepines (PBDs) are a class of sequence-selective DNA minor groove binding natural products, and their strong cytotoxicity makes them a research hotspot as ADC payloads [47]. PBD dimers such as **16a** inhibit a variety of cancer cell lines with subnanomolar IC_50_, which is formed by joining two PBD monomers via their C8-phenol groups with a three-carbon spacer.

A model of **16a** bound to DNA suggested that C7/C7′-methoxy groups in an exposed orientation. A new class of macrocyclic compounds was designed by Donnell et al., which exploits the conformational restriction strategy, joining two groups at the 7/7’ position with an aliphatic linker to improve potency.

At first, they aimed at identifying an appropriately-sized linker for macrocyclization. Modeling of a macrocyclic analog of **16a** (Figure 18) suggested that the length of linkers should fulfill the requirement that it does not distort the spiral binding. Based on this concept, compound **16b** with an 8-carbon chain was obtained, which had substantial improvements in potency.

C2-substituents were optimized to match the macrocyclic PBD scaffold. Dramatically, the 4-methylpiperazine derivative **16c** achieved an improvement in both solubility and potency, which showed an aqueous solubility of more than 698 mg/mL and an IC_50_ below 100 pM in four of the cell lines.

### 2.12. Design and Synthesis of a New, Conformationally Constrained, Macrocyclic Small-Molecule Inhibitor of STAT3 via ‘Click Chemistry’

The activation of signal transducer and activator of transcription 3 (STAT3) is common in patients with advanced cancer, which is dormant in normal cells. Studies have confirmed that the continuous activation of STAT3 can promote tumor cell proliferation, inhibit tumor cell apoptosis, and promote cancer.

In 2007, Chen et al. reported a conformationally constrained macrocyclic peptidomimetic **17b** as a potent STAT3 inhibitor [48]. A small-molecule antagonist that can block the dimerization of STAT3 is a very attractive therapeutic approach for cancers.

In the beginning, the researchers found that in the X-ray co-crystal structure of the peptide segment of **17a** with the dimeric STAT3, the side chains of the two lysines are exposed to solvent. Subsequently, the new STAT3 inhibitor **17b** was designed by cyclizing these two lysine residues to form a macrocyclic ring (Figure 19). The predicted interaction mode of compound **17b** with the dimeric STAT3 was similar with that of **17a**, which enhances the interaction with amino acids.

In the FP-based assay, compound **17b** is three times more potent than the lead peptide **17a**, showing a K_i_ value of 7.3 μM. Modeling showed that compound **17b** interacts with STAT3 in a similar fashion as the linear peptide 1. In summary, macrocyclic peptidomimetics **17b** is a promising initial lead compound for further optimization to obtain potent, cell-permeable, small-molecule inhibitors of STAT3.

### 2.13. Structure-Based Design of a Macrocyclic PROTAC

Since the discovery of targeted protein degradation, the first macrocyclic PROTAC was designed by adding a second cyclizing linker to the BET degrader MZ1 in 2019.

PROTACs are composed of a ligand for a target protein, a flexible linker and a ligand for an E3 ubiquitin ligase, which can degrade target proteins by forming stable ternary complexes. Studies have shown that protein-protein interactions (PPIs) between the ligase and the target can lead to the formation of stable ternary complexes.

The co-crystal structure of a ternary complex usually can provide more information for structure-based drug design. Based on the crystal structure of the PROTAC MZ1 in complex with the E3 ligase VHL and its target Brd4 bromodomain (Figure 20A), researchers found that the VHL ligand VH032 and BET inhibitor JQ1 (the two ligand moieties of MZ1) lay in close spatial proximity. Therefore, macrocyclic PROTACs were designed to lock the PROTAC conformation in the bound state and reduce the energetic penalty [49,50].

In order to retain the original bonding mode and maintain the induced PPIs contributing to the formation of stable and cooperative ternary complex [51,52], two vectors attaching a phenolic of the VHL ligand to the first PEG unit of the MZ1 linker were investigated (vectors A and B, Figure 20B).

Vector B was considered to be more suitable by studying computationally the potential energy strain introduced by alkylation in vectors A and B in the JQ1-amide. Subsequently, modelling studies showed that a linker comprising 3 PEG units (compound **18**) could be well accommodated in the cavity between the proteins. Crucially, the simulations showed that macrocyclization was compatible with the water-mediated interaction of the JQ1-amide with N433BRD4 (BD2), the H-bond of the oxygen atom in the first PEG unit in MZ1 with H437BRD4 (BD2), and the H-bond with Y98VHL (Figure 20).

Biophysical studies revealed that the discrimination between the first and the second bromodomains of BET proteins was strengthened in compound **18**. Moreover, compound **18** showed similar cellular activity with MZ1, despite a 12-fold loss of binary binding affinity for Brd4 appearing. Based on these findings, macrocyclization may be an effective strategy to enhance PROTAC degradation potency and selectivity between homologous targets.

### 2.14. Structure-Based Design of GLS1 Macrocyclic Inhibitors Targeting Allosteric Binding Site

Glutaminase 1 (GLS1), the first key enzyme that catalyzes glutamine to glutamate in glutamine metabolism, has become a potential therapeutic target for treatment of malignant tumors [53,54,55]. In 2021, Xu et al. reported a conformationally constrained GLS1 macrocyclic inhibitor **64b** targeting the allosteric binding site [56]. Based on the published X-ray crystal structures (PDB: 5HL1) of GLS1 as a complex with **64a** (CB-839), which is a known clinical GLS1 allosteric inhibitor, the bound structures of these U-shaped binders underscored the proximity of heteroaromatic tail at both ends, with the distance between the terminal atoms being about 10.58 Å (Figure 21A). A few amino acid residues filled the space between these two heteroaromatic rings, providing a chemical space for the introduction of cyclic chains. Based on the scaffold of **64a**, a series of macrocyclic GLS1 inhibitors were designed and synthesized by linking the terminal heteroaromatic rings. These macrocycles are expected to reinforce the binding conformation to increase the GLS1 inhibitory activity. Compound **64b**, containing 29 ring atoms, exhibited the strongest GLS1 inhibitory activity (IC_50_ = 6 nM), which was more potent than that of **64a** (IC_50_ = 22 nM). Surface plasmon resonance (SPR) analysis indicated that **64b** exhibited robust GLS1 binding affinity (*K*_d_ = 24 nM), which was also stronger than that of **64a** (*K*_d_ = 106 nM). Moreover, **64b** exhibited both preferable antiproliferative activity against HCT116 cells (IC_50_ = 81 nM) and MDA-MB-436 cells (IC_50_ = 340 nM), comparable with that of **64a** (IC_50_ of 110 nM and 430 nM, respectively) [56].

By overlaying the crystal structure of **64a** and **64b** bound to GLS1 (Figure 21B,C), we found that **64b** binds to the same pocket where **64a** binds. This observation indicates that the binding mode of **64b** is similar to that of **64a**, which induces key residues conformational changes of the GLS1 protein to stabilize an open and inactive conformation of the catalytic site. Finally, the effect of glutaminase inhibition on the levels of intracellular metabolites induced by **64b** was also evaluated in HCT116 cancer cells. The results demonstrated that compound **64b** could potently block glutamine metabolism (Figure 21D). Further insights could be provided into the rational design of next-generation GLS1 inhibitors based on the macrocyclic scaffold [56].

### 2.15. Omipalisib Inspired Macrocycles As Dual PI3K/mTOR Inhibitors and Triple PI3K/mTOR/PIM-1 Inhibitors

Activation of the phosphatidylinositol 3-kinase (PI3K)/mammalian target of rapamycin (mTOR) signaling pathway occurs frequently in a wide range of human cancers and is a main driver of cell growth, proliferation, survival, and chemoresistance of cancer cells. In 2021, Pastor et al. reported efforts on the exploration of novel small molecule macrocycles as dual PI3K/mTOR inhibitors [57]. The known clinical PI3K/mTOR inhibitor **65a** (Omipalisib) was selected as a chemical starting point for the discovery and design.

Through co-crystal complex analysis and docking, it was found that the quinoline nitrogen atom of **65a** formed a key hydrogen bond interaction with Val882 in the hinge area. The pyridosulfonamide formed hydrogen bonding interactions with Lys833. Di-fluorophenyl and pyridazine were located in the solvent-exposed region, which can be linked into macrocyclic molecules with suitable chains to fix the dominant conformation and improve the activity. Compound **65b** exhibited potent both PI3K-a and mTOR activities (IC_50_ = 0.8 and 3.3 nM). The inhibitory activity against p-AKT of **65b** was better than the reference compound **65a** (IC_50_: 0.17 vs. 0.6 nM). Moreover, **65b** showed good pharmacokinetic properties with high oral bioavailability (F = 85%), and **65b** displayed excellent profiles with negligible inhibition of the CYP450 enzymes and none binding against h-ERG (IC_50_ > 30 μM), indicating the potential safety of **65b** [57]. Finally, **65b** demonstrated nanomolar antiproliferative activities with Growth Inhibition 50 (GI_50_) values below 200 nM across the 14 cancer cell lines tested (Figure 22). These results open a new avenue for the discovery and design of novel potent and selective PI3K/mTOR dual compounds [57].

Moreover, Pastor found that a macrocyclization strategy achieved the desired polypharmacological profile, starting from an “open precursor” without said combined activity [58]. Compounds **65c**–**65e**, as open precursors, were inactive against PIM-1 (IC_50_ > 10^4^ nM). However, **65f** displayed a potent nanomolar PIM-1 inhibition (IC_50_ = 22.27 nM). The contribution of macrocyclization to the PIM-1 activity of **65f** was outstanding and unexpected. Compound **65g** was a F-derivative of **65f**. Compounds **65f** and **65g** both displayed robust inhibitory activity against PI3K mTOR, and PIM-1 at the nanomolar level. Both **65f** and **65g** were orally bioavailable with absorbed fractions of 35% and 124%, respectively (Figure 23). Because of its more balanced triple PI3K/mTOR/PIM inhibition and pharmacokinetics, compound **65g** (**IBL-302**, **AUM302**) was prioritized for further development. Currently, **65g** is in late preclinical development and undergoing IND-enabling studies [58]. To our knowledge, this is the first reported case of a macrocyclic compound which enables a pursued polypharmacology not available in its non-macrocyclic precursor. This strategy could be applied by medicinal chemists to other bioactive compounds to enrich their target profile and efficacy.

## 3. Synthesis of Macrocyclic Compounds

One of the important challenges in the discovery of macrocyclic drugs is the difficulty in synthesizing such structures, and it is more difficult to obtain a series of molecules to elucidate SAR or build screening libraries. Many feasible synthetic routes have been successfully developed, mainly including amide and ester coupling, nucleophilic substitution reaction, ring-closing metathesis (RCM), and “click” chemistry cycloadditions.

### 3.1. Synthesis of Macrocyclic Compounds Using Coupling Reaction

#### 3.1.1. Synthesis of HSP90 Macrocyclic Inhibitors Using Buchwald-Hartwig Coupling Reaction

Hsp90 macrocyclic inhibitors were designed and synthesized by Christoph W. Zapf et al. in 2011 [20]. They used the Buchwald-Hartwig coupling reaction for macrocyclization and synthesized a series of HSP90 macrocyclic inhibitors. Figure 1 shows the synthetic route for compound **10g**. After dihydroxylation, diol cleavage and oxidation under suitable conditions, homoallyl tetrahydroindolinone **19** is advanced to the aldehyde-based intermediate **20**. This is then appropriately converted to the desired primary or secondary amine such as intermediate **21** as shown. Acylation protection using alanine **22** followed by deprotection, Buchwald cyclization and hydrolysis results in macrocycle **10g**.

#### 3.1.2. Synthesis of BET Macrocyclic Inhibitors using Sonogashira Coupling Reaction

In 2017, Le Wang et al. selected the substitution reaction of alcohol hydroxyl and halogen to complete cyclization when designing macrocyclic BET inhibitors [14]. Figure 2 shows the synthetic route for compound **5e**. Sonogashira coupling of compound **24** followed by hydrogenation resulted in compound **26**. Then, the reaction of compound **27** with compound **26** produced pyridone **28**, and palladium-catalyzed borylation, which resulted in compound **29**. Pyridine **31** was obtained by reacting **29** with three 3-bromo-2-chloro-5-subtituted pyridines, the reaction of which with boron tribromide resulted in compound **32**. Intramolecular nucleophilic substitution mediated by Cs_2_CO_3_ in acetonitrile successfully converted compound **32** to the macrocycle compound **5e** with excellent yields.

### 3.2. Synthesis of CDK Macrocyclic Inhibitors Using Nucleophilic Substitution Reaction

Kawanishi et al. discovered a series of macrocyclic CDK1,2,4,6 inhibitors in 2006 [45]. After the modification and screening of the obtained compounds, **16g**, a superior compound, was finally obtained. The synthesis of the final product utilizes the nucleophilic substitution reaction between the sulfonyl group and the hydroxyl group. The synthetic route is shown in Figure 3. Starting from the starting material **33**, the intermediate **39** is obtained through a series of substitution, condensation, catalysis and other reactions. The alcohol is mesylated and then aminated with (3R, 5R)-5-methylpyrrolidine-3-ol8 to give compound **40**. Compound **40** was mesylated, and then the MOM group was removed with TFA at room temperature to obtain compound **41**. Finally, intermediate **41** undergoes a nucleophilic substitution reaction under alkaline conditions to obtain the final product **16g**.

### 3.3. Synthesis of CK2 Macrocyclic Inhibitors using a Condensation Reaction

Condensation is an important method for the synthesis of macrocyclic compounds. In 2008, Nie et al. synthesized a novel class of macrocyclic CK2 inhibitors through a condensation reaction (Figure 4) [43]. Dicyanopentane was selected as the key intermediate for the construction of macrocycles, which was successively treated with NaH, ethylformate, then cyclized with hydrazine to provide 4-substituted amino pyrazole **43**. Pyrazolotriazine **44** was obtained by reacting **43** with thioisocyanate in refluxing EtOAc, followed by cyclization in the presence of a base. After benzoylation, chlorination, and replacement of C4-chloro with cyclopropylamine, compound **47** was obtained. After a three-step reaction, compound **49** is converted to compound **50**. The intra-molecular cyclization was readily achieved by treatment with HATU in the presence of base in NMP to provide macrocyclic pyrazolotriazine **15b**.

### 3.4. Synthesis of Chk1 and HDAC Macrocyclic Inhibitors Using Ring-Closing Metathesis (RCM) Reaction

Zhi-Fu Tao et al. used the RCM reaction in the cyclization of a ChK1 inhibitor, which is the most widely used reaction for the synthesis of macrocyclic structures [33]. For subsequent RCM reaction, terminal olefins on both sides were constructed. After commercially available aminophenol **51** is alkylated with allyl bromide, it is treated with phosgene in toluene to form isocyanate **53**. Aminopyrazinyl ether **55**, prepared through the nucleophilic displacement of chloride from **54** by 3-butenol, was coupled to **53** to give the key intermediate **56** (Figure 5). Subsequently, the macrocyclization of **56** was carried out under the catalysis of Grubbs catalysts, and the reduction of cyclic olefin **57** provided 11e.

Similarly, macrocyclic HDAC inhibitors were synthesized through the RCM reaction which was catalyzed by Hoveyda-Grubbs’ catalysts. The cleavage and reconstruction of carbon-carbon bonds were completed during the RCM reaction, and two carbon atoms were removed to form ethylene gas in the formation of a new carbon-carbon double bond. On the basis of the pioneering work of Grubbs, many catalysts have been discovered one after another [59]. Currently, commonly used catalysts include first generation Grubbs catalysts, second generation Grubbs catalysts and Hoveyda-Grubbs catalysts. Second generation Grubbs catalysts are the most popular choice and must be used under strictly anhydrous and oxygen-free conditions [60,61]. The RCM reaction can adapt to different ring sizes, and the reaction conditions are mild and well tolerated by other chemical functionalities. However, the product of the reaction is usually a mixture of geometric isomers of olefins, and the yield is susceptible to substrate type, space factors and heteroatoms. Moreover, removal of the catalyst to pharmaceutically acceptable trace levels can be technically problematic [6].

### 3.5. Synthesis of STAT3 Macrocyclic Inhibitors Using “Click Chemistry” Cycloaddition Reaction

Another viable methodology is the “click chemistry” cycloaddition reaction between alkynes and azides. In 2006, Zhu et al. reported a straightforward route to macrocycles via a tandem three component reaction with an azide-alkyne [3 + 2] cycloaddition [62]. This research shows that the formation of 14-, 15-, and 16-membered ring macrocycles is feasible, and the yield of cyclization is 24–76%. Even though this methodology can suffer from the significant formation of dimers or other oligomeric products, which still have the potential to be extended to larger numbers of compounds. In addition, a triazole must be accepted in the final product, and click chemistry is particularly tolerant of functional groups, being one of the few biocompatible ligation techniques. In the construction of a macrocyclic STAT3 inhibitor, key intermediate **63** was produced via “click chemistry” in the presence of CuSO_4_·5H_2_O/sodium ascorbate (Figure 6). Compound **60**, produced by condensation of Boc-L-6-hydroxynorleucine **58** with O-t-butyl-L-threonine methyl ester **59**, was successively treated with methanesulfonyl chloride, sodium azide and LiOH/HCl to produce the azide **61**. The condensation of which with (S)-propargylglycine methyl ester furnished compound **62**, and this was converted in 80% yield into the key intermediate **63**.

## 4. Conclusions and Perspective

Synthetic macrocyclic compounds have great chemical diversity, limited only by our imagination. As summarized in the current targets, synthetic macrocyclic small-molecule drugs are still rare, but they have already proven their value as antitumor agents. Some macrocycles have been approved for marketing or have entered clinical trials (Table 6). **2d** (**Lorlatinib**), a highly potent and selective ALK/ROS1 inhibitor, was approved for the treatment of NSCLC in 2018 [63]. Compounds **3c** and **3d** have the same scaffold structure as 2-aminopyrimidine-based macrocycles. They all exhibited robust inhibitory activity against JAK2, while target selectivity was slightly different. **3c** (**Pacritinib**) is a potent JAK2/FLT3 inhibitor, which was approved by FDA for the treatment of patients with myelofibrosis and severe thrombocytopenia in 2022 [64,65]. **3d** (**Zotiraciclib**) targeting JAK2/FLT3/CDK2 is in phase 1/2 clinical trials for treatment of leukemia and multiple myeloma [66]. **67** (**Temsirolimus**) and **68** (**Everolimus**) belong to structural analogs of **66** (**Sirolimus**, also named **rapamycin**). They are all mTOR inhibitors. **67** (**Temsirolimus**) and **68** (**Everolimus**) were approved for the treatment of renal cell cancer [67,68]. **66** (**Sirolimus**) was approved to prevent transplant rejection after kidney transplantation [69]. **69** (**E6201**) as an MEK1/FLT3 inhibitor has entered into a clinical phase I study for treatment of malignant melanoma [70,71,72]. **70** (**Repotrecitinib**) and **71** (**Selitrecitinib**, **LOXO-195**) are robust TRKA/B/C inhibitors. Moreover, **70** (**Repotrecitinib**) additionally inhibited ROS1/ALK. The two macrocyclic compounds (**70** and **71**), which were designed to overcome the resistance to first generation of TRK inhibitors [73], are in Phase1/2 clinical trials for treatment of advanced solid tumors. **72** (**JNJ-26483327**) was entered into a phase 1 clinical trial in 2008 as a multitargeted tyrosine kinase inhibitor which was developed for the treatment of advanced and/or refractory solid malignancies [74,75].

Synthetic macrocyclic small-molecule drugs, an excellent way to lock out alternative conformations, are being driven by multiple factors. In a conformationally pre-organized ring structure, a macrocycle provides diverse functionality and stereochemical complexity, which can result in high affinity and selectivity for protein targets. As the standard small molecule oral drug property rules (e.g., Lipinski’s rule of five 6) are somewhat relaxed for macrocyclic drugs, this represents an opportunity to find and develop small molecule drugs with properties well outside conventional ideas. Macrocyclic molecules tend to have better targets and cellular activities than their acyclic precursor compounds. This is because the macrocyclic molecule fixes a dominant conformation, which reduces the loss of entropy when interacting with the protein and is beneficial to improve the affinity with the protein due to the entropy-enthalpy compensation effect. In addition, the newly added linker sometimes forms new interactions with the protein, thereby further enhancing the activity of the ligand. This feature of macrocyclic molecules is mostly used in the development of kinase inhibitors and can be used to design macrocyclic drugs against kinase inhibitor resistance. In addition, another advantage of macrocyclic molecules is that they have good physicochemical properties, such as membrane permeability. Macrocyclic molecules are very suitable for the development of central nervous system (CNS) drugs, which need to penetrate the blood-brain barrier (BBB). This is because the macrocyclic molecular structure is semi-rigid, relatively lipophilic, and easily penetrates the BBB. Moreover, advances in synthetic chemistry technology have led to many reactions, including the Buchwald–Hartwig coupling reaction, Heck reaction, RCM reaction, and click chemistry, successfully applied to the synthesis of macrocyclic compounds.

Cancer is one of the deadliest diseases in humans. Cancer cells have complex mechanisms of proliferation, invasion, and metastasis, which create many difficulties for the discovery of targeted therapies. In addition, existing targeted therapies need to be continuously upgraded to cope with emerging resistance. The potent inhibitory potency of ALK inhibitor **2d** (**Lorlatinib**) and EGFR inhibitor BI-4020 (**4d**) demonstrates the potential of macrocyclic compounds to combat resistance. Macrocyclic inhibitors of the other 10 targets also show their own advantages in inhibitory efficacy, good solubility, metabolic stability and bioavailability. In general, macrocycles are ideal to tackle “difficult” targets and overcome molecular property challenges in small-molecule drug discovery.

## Data Availability

Not applicable.

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
