# Peer review of "An Insight into the Medicinal Chemistry Perspective of Macrocyclic Derivatives with Antitumor Activity: A Systematic Review"

_molecules, 2022, doi:10.3390/molecules27092837_

Round 1
Reviewer 1 Report
In this paper, Liang et al. summarized characters and synthetic routes of anticancer macrocyclic molecules as a systematic review. This theme is interesting, and basically this manuscript is well written. The manuscript can be accepted after consideration of the following points.
- In introduction, lines 27-31 are important to explain significance of macrocycles. Especially expression “reducing the major entropy loss” is important point, but it is difficult for many medicinal chemists to understand clearly. Please explain using easily understandable words.
- Line 32, Does “Compared with high molecular compounds” mean “Compared with low molecular compounds”? Furthermore, it cannot be understood why cyclization increases the molecular weight (line 33). It is better to write various characters of macrocyclic molecules (lines 33-34) in “4. Conclusion and perspective”. Or readers can not be understood why these characters are expressed in the macrocyclic molecules.
- Throughout the manuscript, styles of figures and tables should be unified. For example, styles of Fig. 1C, Fig. 3, Fig. 6 are very different. Sizes of structures should be unified. It looks as if the figures are jumbled.
- Line 92, It can not be understood the meaning of “a reduced surface area”. It looks to be an important point. Please explain easily.
- Figure 3, It is better to add activity of 3a in the figure.
- In “4. Conclusion and perspective”, the summary of characters of macrocyclic molecules and why these are expressed by cyclization should be written.
Reviewer 2 Report
This review focuses on the antitumor properties of thirteen representative macrocycles via different targets, which gives us a deep insight into the chemical modification of macrocyclic compounds associated with their pharmacological properties.
After reviewing the manuscript, I found the majority of the cited references were published ten years ago. How about the research in recent years? For researchers, they pay more attentions on recent research study on the selected compounds and their analogues.
And another query is that almost the cited research papers selected from two journals JMC & BMCL. How about the research work published on other journals?
In the introduction part, it is better to give a brief introduction of ten macrocyclic drugs used to treat cancers as mentioned in the abstract, which could be an interesting reading guide.
There are a lot of data in Tables and Figures. However, due to the insufficient citations in the main text especially the front part, it is difficult to find the origins of these data. Please add references in the appropriate paragraphs or table footnotes or figure captions.
Aside, improvements are required for this manuscript. Some of them are listed as followings.
- There is an typo in the reference [1] title. Please revise ‘ligand ability’ as ‘ligandability’.
- The reference [10] title is incomplete. It is ‘Discovery of (10R)-7-Amino-12-fluoro-2,10,16-trimethyl-15-oxo-10,15,16,17-tetrahydro-2H-8,4-(metheno)pyrazolo[4,3-h][2,5,11]-benzoxadiazacyclotetradecine-3-carbonitrile (PF-06463922), a Macrocyclic Inhibitor of Anaplastic Lymphoma Kinase (ALK) and c-ros Oncogene 1 (ROS1) with Preclinical Brain Exposure and Broad-Spectrum Potency against ALK-Resistant Mutations’.
- Please give a explanation for the abbreviation ‘MDR BA/AB’ when it appeared for the first time in this manuscript.
- The number ‘1’ was missing in the abbreviation on Page 2 Line 78: ‘P-glycoprotein 1 (Pgp)’ →‘P-glycoprotein 1 (Pgp 1)’.
- According to the table body, it is better to modify the table caption as ‘Table 1. Biological and physicochemical profiles of macrocyclic ALK inhibitors’. And please modify the style according to the Instructions for Authors or the recently published papers.
- Please revise ‘with compound 2b’ as ‘to those of compound 2b’ on Page 4 Line 106.
- ‘residues which was’→ ‘residues which were’ on Page 4 Line 111.
- ‘TrxB’ → ‘TrkB’ on Page 4 Lines 117 & 121.
- ‘terminal OH group (OH)’ → ‘terminal hydroxyl group (OH)’ on Page 4 Line 118.
Indeed, distances were measured between the methyl carbon atom of 2c and the terminal atom oxygen in Figure 2A, and between the nitrile nitrogen atom of 2d and the terminal atom oxygen in Figure 2B.
- ‘penalties’ → ‘penalty’ on Page 4 Line 124.
- ‘CNS’ is the abbreviation of ‘central nervous system’, and ‘CNS’ appears in the main text frequently, therefore please delete ‘central nervous system’ but keep ‘CNS’ in the sentence on Page 5 Line 134.
- Please add the full name ‘nonsmall cell lung cancer’ before the abbreviation ‘NSCLC’ on Page 5 Line 137.
- ‘myeloproliferative tumors (MPNs)’ → ‘myeloproliferative neoplasms (MPNs)’ on Page 6 Line 155.
- Please revise the sentence as ‘Macrocyclic compound 3e has significant inhibitory activities on JAK2, FLT3 and TYK2, but its inhibitory activities on JAK1, JAK3, and CDK2 were significantly weaker…’ on Page 6 Lines 163–165.
- Page 7 Lines 189 & 190: As shown in Figure 4, the R1 substituent on the C ring is substituted with a pyrrolidine carbon chain, not R2.
- ‘up to day 55’ →‘up to 55 days’ on Page 8 Line 205.
- ‘ASP86’ → ‘Asp86’ on Page 8 Line 225.
- ‘biological activity evaluations’ →‘biological evaluations’ on Page 8 Line 230.
- ‘potent potency’ →‘potent inhibitory’ on Page 8 Line 231.
- ‘EGFRwt’ → ‘EGFRwt’ on Page 9 Line 260.
- For the notes below the structures 4c and 4d in Figure 6, ‘EGFRdel19 T790M C797S’ → ‘EGFRdel19 T790M C797S’.
- ‘R1-position’ → ‘R1-position’ on Page 10 Line 283.
- ‘R3-position’ → ‘R3-position’ on Page 10 Line 286.
- The reference [14] title is incomplete. It is ‘Fragment-Based, Structure-Enabled Discovery of Novel Pyridones and Pyridone Macrocycles as Potent Bromodomain and Extra-Terminal Domain (BET) Family Bromodomain Inhibitors’.
- The original journal information for reference [16] is ‘Bioorg. Med. Chem. Lett., 2012, 22, 1136-1141.’
- The original article information for reference [21] is ‘Heat shock protein 90: inhibitors in clinical trials. J. Med. Chem., 2010, 53, 3-17.’
- ‘G2’ → ‘G2’ on Page 16 Line 417.
- ‘the modification of the C6′-position in the pyrazinyl ring’ on Page 17 Line 426, whereas the position is labeled as ‘6’ not ‘6′’ in the structures of 11c and 11d in Figure 13.
- ‘The X-ray cocrystal structure of 11e (Figure 14B) showed that the carbon chain makes favorable van der Waals contacts with Glu91, Asn135, Leu137, Glu134, and Ser147.’ Indeed, none of these residues were shown in Figure 14B. By the way, one methyl bonded to the N atom in the structure of 11d was lost seemingly in Figure 14A.
- ‘G2max M’ → ‘G2/M’ on Page 18 Line 451.
- The numbering in the pyrazinyl ring in the structure shown in Table 2 was wrong. ‘6’ should be revised as ‘5′’.
- The year for the publication [37] is ‘2010’, not ‘2009’.
- Two words were missing in the title of the publication [38]. It is ‘Histone deacetylase inhibitors: from bench to clinic’, not ‘2009’.
- Some data need to be revised in Table 3 according to the reference [42]. For example, ‘27’ → ‘28’, ‘15’ → ‘16’, ‘52.10’ → ‘52’.
- According to the reference [42], the unit for the data ‘0.84 μM’ recorded on Page 20 Line 480 should be revised as ‘nM’.
- According to the reference [42], the unit for the data ‘0.40 μM’ recorded on Page 20 Line 487 should be revised as ‘nM’.
- What do R substituent represents for compounds 14c–14j in Figure 16A?
- ‘x-ray’ → ‘X-ray’ on Page 22 Line 553.
- What is the structure of compound 18? Please show it in Figure 20.
- ‘sonogashira’ →‘Sonogashira’ on Page 26 Line 668.
- The original journal information for reference [53] is ‘J. Am. Chem. Soc., 1996, 118, 9606–9614.’
Round 2
Reviewer 1 Report
The revised manuscript is considered to reach the level of acceptance.
Reviewer 2 Report
The comments has been addressed and the paper is revised accordingly. No further suggestion from me. Now it is a nice paper, and I recommend it for publication.